# Matrix-Assisted Laser Desorption/Ionization Time of Flight Mass Spectrometry (MALDI-TOF MS) Analysis for the Identification of Pathogenic Microorganisms: A Review

**DOI:** 10.3390/microorganisms9071536

**Published:** 2021-07-19

**Authors:** Xin-Fei Chen, Xin Hou, Meng Xiao, Li Zhang, Jing-Wei Cheng, Meng-Lan Zhou, Jing-Jing Huang, Jing-Jia Zhang, Ying-Chun Xu, Po-Ren Hsueh

**Affiliations:** 1Department of Laboratory Medicine, State Key Laboratory of Complex Severe and Rare Diseases, Peking Union Medical College Hospital, Chinese Academy of Medical Science and Peking Union Medical College, Beijing 100730, China; chenxinfei29@163.com (X.-F.C.); houxinyffs@163.com (X.H.); cjtcxiaomeng@aliyun.com (M.X.); zhangli19870711@163.com (L.Z.); mumuxi529@139.com (M.-L.Z.); hjj_6620@163.com (J.-J.H.); zczjj2009@126.com (J.-J.Z.); 2Graduate School, Chinese Academy of Medical Sciences and Peking Union Medical College, Beijing 100730, China; 3Beijing Key Laboratory for Mechanisms Research and Precision Diagnosis of Invasive Fungal Diseases, Beijing 100730, China; 4Center of Clinical Laboratory, Beijing Friendship Hospital, Capital Medical University, Beijing 100053, China; zzujingwei@163.com; 5Departments of Laboratory Medicine and Internal Medicine, China Medical University Hospital, School of Medicine, China Medical University, Taichung 40447, Taiwan; hsporen@gmail.com; 6Departments of Laboratory Medicine and Internal Medicine, National Taiwan University Hospital, National Taiwan University College of Medicine, Taipei 100, Taiwan

**Keywords:** matrix-assisted laser desorption/ionization–time-of-flight mass spectrometry (MALDI-TOF MS), pathogenic microorganism identification

## Abstract

Matrix-assisted laser desorption ionization time-of-flight mass spectrometry (MALDI-TOF MS) has been used in the field of clinical microbiology since 2010. Compared with the traditional technique of biochemical identification, MALDI-TOF MS has many advantages, including convenience, speed, accuracy, and low cost. The accuracy and speed of identification using MALDI-TOF MS have been increasing with the development of sample preparation, database enrichment, and algorithm optimization. MALDI-TOF MS has shown promising results in identifying cultured colonies and rapidly detecting samples. MALDI-TOF MS has critical research applications for the rapid detection of highly virulent and drug-resistant pathogens. Here we present a scientific review that evaluates the performance of MALDI-TOF MS in identifying clinical pathogenic microorganisms. MALDI-TOF MS is a promising tool in identifying clinical microorganisms, although some aspects still require improvement.

## 1. Introduction

Matrix-assisted laser desorption ionization time-of-flight mass spectrometry (MALDI-TOF MS) has been applied in the field of clinical microbiology for a dozen years. With the development of new technology and optimization of methods, novel rapid and accurate approaches have been developed to improve identification accuracy. Clinical microbiology laboratories are constantly expanding the application of MALDI-TOF MS in new areas of infectious disease diagnostics. Accurate identification of microbes at the species level in clinical situations is imperative in most cases. Therefore, this review focuses on the efficacy of MALDI-TOF MS in identifying pathogens at the species level in a clinical setting.

MALDI-TOF MS is not limited to identifying strains cultured on solid media but can also directly identify them from the blood culture, cerebrospinal fluid, and urine [1,2,3,4,5]. MALDI-TOF MS identification has been used to identify gram-negative and gram-positive bacteria, aerobes, anaerobes, mycobacteria, *Nocardia*, yeasts, filamentous fungi and viruses [6,7,8,9,10,11,12,13].

MALDI-TOF MS has good identification outcomes for common, clinically isolated bacteria [14,15]. In fungi, the identification of *Candida* spp. and filamentous fungi has made good progress [4,11]. MALDI-TOF MS plays an important role in the identification of viruses, including influenza viruses, hepatitis viruses, and herpesviruses [16,17,18,19]. New studies have shown remarkable progress in the detection of microorganisms by MALDI-TOF MS and the identification of drug-resistant strains [20,21,22,23,24,25].

The present review aimed to summarize the efficacy of MALDI-TOF MS for pathogen identification in clinical microbiology laboratories, including sample preparation and identification performance of microorganisms. In addition, the current problems associated with MALDI-TOF MS in clinical laboratories are also discussed.

## 2. MALDI-TOF MS-Based Microorganism Identification and Sample Preparation

### 2.1. MALDI-TOF MS Basic Overview

During MALDI-TOF MS detection, the microbial samples, with or without a preceding extraction step, and the matrix are added to the target and dry to form crystals. The sample and matrix absorb energy and vaporize into a positively charged cloud when the plate is exposed to laser irradiation. These ionized proteins are accelerated in the flight tube under an electrostatic field. The speed at which ions fly to the ion detector depends on the mass-to-charge ratio (*m/z*). The value of *m/z* is shown on the *x*-axis, while the signal strength is shown on the *y*-axis of the mass spectrum. The height of the peak in the mass spectrum is related to the abundance of a protein. MALDI-TOF MS can analyze the positively charged proteins of 2000–30,000 Da for the identification of microorganisms. Different MALDI-TOF MS systems can detect ionized proteins in different ranges: the Shimadzu system can detect ionized proteins smaller than 30 kD, while the Bruker system can detect ionized proteins smaller than 20 kD. Therefore, the spectra (in clinical settings) only included a small part of the entire spectra, composed of the proteins of the microorganisms, and subsequently were compared with the database [26].

### 2.2. Sample Preparation

The majority of bacterial and yeast samples in clinical laboratories are separated on a solid agar medium, following which, an appropriate colony is selected and smeared directly on a target plate as a thin film. However, *Mycobacterium* spp., *Nocardia* spp., and filamentous fungi have complex wall structures, resulting in poor protein extraction with the matrix process. The formic acid extraction method (on-plate) can improve extraction efficiency. Unlike bacteria and fungi, which can be cultured on solid media, viruses can only be cultured in living cells. In addition, due to different detection principles, the preparation for MALDI-TOF MS-based virus identification is nucleic acid extraction. With the recent development of MALDI-TOF MS, there has been an improved study of direct sample identification [27,28,29,30].

Positive blood bottles and urine samples may require further processing to retain pathogens. In addition, workflow and preparation process for different samples need to be established to improve MALDI-TOF MS identification.

#### 2.2.1. General Preparation with Laboratory Culture Strains

Identification of culture strains is a routine clinical laboratory technique. For most bacterial and yeast strains, direct deposition is an easy and rapid method [31]. This method includes two steps: first, the deposition of a single colony on the target and fixing it with a matrix. On-target extraction has a good effect on yeast and some difficult-to-extract bacteria [32,33,34]. The smeared samples were first treated with formic acid. After air-drying, the samples were dropped onto the matrix. If the direct deposit or on-target extraction methods cannot identify the microorganisms in strains with a thick wall or capsule layer, tube extraction is recommended [35]. This method uses ethanol treatment followed by formic acid and acetonitrile to extract the proteins. An easy and effective preparation method is preferable for routine identification. Protein extraction from single colonies on a solid medium must be optimized [36].

To date, two primary methods have been reported for virus detection by MALDI-TOF MS. One is to detect virus proteins via the analysis of protein peaks by MALDI-TOF MS [16]. This process is similar to the preparation of bacteria and fungi. The second is to detect amplified viral nucleic acids. General preparation procedures can be roughly divided into two stages: (1) nucleic acid extraction and PCR amplification, and (2) single base extension of quality probes [19,37,38].

#### 2.2.2. Liquid Samples

MALDI-TOF MS has been applied in clinical liquid samples, including blood culture, urine, cerebrospinal fluid (CSF), and various other body fluids; however, it is limited by the low number of pathogens and abundance of human cells [39]. Bacteria with a load of 10^5^–10^7^ colony-forming units (CFUs) can be identified by MALDI-TOF MS [40]. For positive blood cultures, the sample needs to be concentrated and purified before identification. Various protocols have been reported to identify microorganisms present in positive blood culture broth accurately. Home-brew protocols were developed as follows: 1. Stepwise differential sedimentation of blood cells and microorganisms; 2. Centrifugation and lysis procedures; 3. Lysis-vacuum filtration; and 4. Centrifugation and membrane filtration techniques [20]. These processes may be performed by differential centrifugation and washing, selective lysis of blood cells, serum separator tubes, or filtration. Recently, a short incubation period based on positive blood culture has been used to improve the identification rate by using MALDI-TOF MS. These techniques measure the positive blood culture broth on a solid medium incubated for approximately 4–6 h and identify the colonies by using MALDI-TOF MS [27,41]. Although this short incubation method is tedious, it can benefit the diagnosis and treatment of patients with bloodstream infections. However, the lack of standardized protocols, the use of different software for mass analysis, and different blood culture bottles make it difficult to compare the performances of the different methods.

In addition to the home-brew methods, there are three major commercial kits: the Sepsityper^®^ kit (Bruker Daltonics GmbH, Bremen, Germany), the VITEK MS blood culture kit (bioMerieux, Marcy-l’Étoile, France), and the BACpro^®^ II kit (Nittobo Medical Coo, Tokyo, Japan) [20,42,43]. Currently, the cost of commercial kits is high, and it must be reduced.

Pathogens in samples of positive urine routine tests detected by flow cytometry can be directly identified using MALDI-TOF MS [44]. Pre-treatment of the urine sample aims to remove human cells via centrifugation. The bacterial pellet is treated with formic acid, followed by MALDI-TOF MS. This method is of great significance for the rapid identification of pathogens in CSF samples. Previous studies have shown that positive smear microscopic examination samples treated after pathogen enrichment and centrifugal stratification to remove human cells can be used for MALDI-TOF MS identification [45].

#### 2.2.3. *Mycobacterium* and *Nocardia* spp.

The cell walls of *Mycobacterium* spp. and *Nocardia* spp. are thick. As a result, formic acid and matrix have limited effects on their cell walls. Therefore, the method of on-plate extraction does not perform well in identifying *Mycobacterium* spp. and *Nocardia* spp. Hence, the tube extraction method can effectively enhance the identification of the *Mycobacterium* spp. and *Nocardia* spp. The key to in-tube extraction is to use physical or chemical methods to destroy the cell wall and then use formic acid and acetonitrile to extract the proteins. The extraction methods for mycobacteria mainly include commercial-kit based extraction, with VITEK MS *Mycobacterium*/*Nocardia* kit being the commonly used commercial kit, and some protocols including sonication, bead beating, and vortexing in the presence of silica beads [46,47,48]. The sample preparation of *Nocardia* spp. by MALDI-TOF MS is similar to that of *Mycobacterium* spp. The identification efficacy of the extraction method has been reported to be better than that of the direct smear method [49]. The mycobacterium genus contains various species, some of which are highly pathogenic and must be inactivated before identification. The inactivation methods are different in the routine operation guidelines of Bruker Biotyper and VITEK MS. Bruker Biotyper uses the thermal inactivation method, whereas VITEK MS uses mechanical crushing with alcohol as the inactivation method. The samples were then treated with formic acid and acetonitrile, the extract was dripped onto the target plate, and the matrix was added after drying [50].

#### 2.2.4. Filamentous Fungi

Currently, the identification protocol for filamentous fungi uses the home-brew method in the laboratory [11,51]. The conventional matrix-assisted dissociation effect of formic acid and acetonitrile and a matrix in the laboratory is imperfect, and the extraction effect must be improved by physical or enzymatic lysis of the microbial cell wall. The mycelium is extracted by a modified formic acid extraction method. The ethanol-treated mycelium extract was harvested by centrifugation and air-dried. Then the extract was incubated with formic acid and acetonitrile, and finally dropped onto a target [11,52].

## 3. Accuracy for Microbial Identification and Other Application

### 3.1. Identification from Culture Plates

#### 3.1.1. Aerobe Bacteria Identification

MALDI-TOF MS technology shows advantages over traditional laboratory diagnostic technologies. The traditional identification of routine clinical culture bacteria relies on biochemical and microscopic examinations and other methods [53]. MALDI-TOF MS shortens the time required for traditional biochemical identification and improves the accuracy of traditional microscopic examinations [39,54]. During the evaluation and analysis of common clinical isolates, the accuracy of determining the species level of common clinical pathogens was found to be close to 90% (Table 1). In 2009, a study comparing different assays for the identification of 1116 clinically encountered bacteria by using MALDI-TOF MS and biochemical testing systems showed correct species identification in 95.2% of cases [14]. Another study obtained the correct identification rate of 1660 bacterial isolates from 109 different species in 84.1% of cases [55]. In a study by Carbonnelle et al., 94.9% and 83.4% of 296 isolates involving gram-positive and gram-negative bacteria, respectively, were correctly identified at the genus and species levels by MALDI-TOF MS [56]. In 2010, a study demonstrated a high rate of correct identification by MALDI-TOF MS compared to traditional biochemical tests in 99.1% of cases [5]. Several studies have been performed to evaluate the ability of MALDI-TOF MS to identify gram-negative bacterial strains [57,58] and have identified it as a suitable and reliable tool for the identification of gram-negative rods [6,59]. In a more recent study, Faron et al. evaluated the Bruker MALDI Biotyper system to identify gram-negative bacteria in a study including 2263 isolates and correctly identified 98.2% of the species [60]. In another study, correct identification was achieved in 87.83% of cases at the species level and 95.9% at the genus level [61]. Rychert et al. reported a multi-center evaluation of MALDI-TOF MS that correctly identified 92.8% gram-positive pathogens at the species level [62]. In addition, Kassim et al. identified anaerobic gram-positive strains [61]. In conclusion, MALDI-TOF MS has a good identification rate in case of common clinical non-fermentative bacteria, Enterobacteriaceae, *Staphylococcus*, *Streptococcus*, *Enterococcus*, *Corynebacterium*, and HACEK strains [63].

#### 3.1.2. Anaerobic Bacteria Identification

Anaerobic bacteria are a major contributor to the human microbiota, with some colonizing the human body. Several species of anaerobic bacteria are the leading causes of clinical infections leading to death. Thus, the accurate identification of anaerobic bacteria plays an essential role in clinical diagnosis and treatment while avoiding unnecessary medication. MALDI-TOF MS has overcome some of the limitations of biochemical tests, greatly reducing the turnaround time and providing faster results [53,64]. Several studies have evaluated the ability of MALDI-TOF MS in identifying anaerobic bacteria. These studies have shown that the identification rate of MALDI-TOF MS is above 80% (Table 1). MALDI-TOF MS can identify common anaerobes in the clinic, and the accuracy of identification is higher than that of traditional biochemical test methods. Garner et al. showed that MALDI-TOF MS correctly identified 92.5% (602/651) anaerobic bacteria at the genus level and 91.2% (591/651) at the species level [65]. Another study reported that the identification rates of MALDI-TOF MS were 99.3% and 89.1% at the genus and species levels, respectively [7]. Rodríguez-Sánchez et al. showed 97.0% correct identification at the genus level and 85.8% at the species level [66]. In a meta-analysis, Li et al. evaluated the efficacy of MALDI-TOF MS in identification of clinical pathogenic anaerobes, including *Bacteroides* spp., *Lactobacillus* spp., *Parabacteroides* spp., *Clostridium* spp., *Propionibacterium* spp., *Prevotella* spp., *Veillonella* spp., and *Peptostreptococcus* spp. The authors reported that MALDI-TOF MS correctly identified the anaerobes at the genus and species levels at 92% and 84%, respectively. The identification accuracy of *Bacteroides* spp. was the highest, whereas the highest identification errors were for rare species [67]. Wang et al. reported the accuracy of three MALDI-TOF MS systems and the VITEK 2 ANC card to identify 138 clinical *B. fragilis* groups. Overall, 94.2%, 94.2%, 98.6%, and 94.9% were identified to the species level using VITEK MS, Clin-ToF-II MS, Autof MS 1000, and the VITEK 2 ANC card, respectively [3]. Therefore, MALDI-TOF MS is a proven useful method, although some anaerobes whose profiles are similar cannot be identified to the species level [68].

#### 3.1.3. *Mycobacterium* and *Nocardia* Identification

Clinically, *Mycobacterium* is the most important genus of bacteria. The *Mycobacterium tuberculosis* complex causes tuberculosis in humans. Traditionally, the biochemical phenotypic identification of mycobacteria lacks accuracy [69]. Molecular methods are widely used in clinical laboratories because of their high identification accuracy; however, these methods are time-consuming [70]. MALDI-TOF MS overcomes the shortcomings of phenotypic and molecular techniques. Recently, several studies have shown that MALDI-TOF MS is a valuable tool for identifying *Mycobacterium* isolates (Table 1). Previous studies have reported that the identification accuracy of *Mycobacterium* is 62–80%. Wilen et al. used MALDI-TOF MS to identify 157 *Mycobacterium* isolates, of which 89.2% (140/157, Vitek MS v3.0) were correctly identified at the species level [50]. Rodrigues-Sanchez et al. reported that 68.8% (86/125) and 88.8% (111/125) of nontuberculous mycobacteria were correctly identified at the species and genus levels by using MALDI-TOF MS [46]. In addition, Chen et al. reported that MALDI-TOF MS could correctly identify 62.8% (64/102) and 87.3% (89/102) at the species and genus levels [71]. At present, MALDI-TOF MS identification of *M. tuberculosis* can only identify the *M. tuberculosis* complex [8]; for nontuberculous mycobacterium, the detection of *Mycobacterium avium* complex is not sufficient to distinguish between *Mycobacterium intracellulare* and *Mycobacterium chimera* [72]. However, Epperson et al. reported that a novel MALDI Biotyper algorithm might accurately identify *Mycobacterium intracellulare* and *Mycobacterium chimera* [73].

For the identification of *Nocardia* spp., the mass spectra data available for *Nocardia* spp. are limited; thus, a self-built database can improve the identification rate. Blosser et al. conducted a multi-center study using three Nocardia libraries (Bruker, National Institutes of Health, and Ohio State University Libraries) to evaluate the efficacy of the Bruker MS method [74]. The identification efficacy upon using the three databases was significantly better than using a single database in all centers. Therefore, constantly updating the database is an indispensable factor for improving the MALDI-TOF MS identification accuracy rate [74]. A commercial database combined with a self-built database was used, and it resulted in an identification accuracy of 90% [75]. Durand et al. evaluated the performance of VITEK^®^ MS (V3.0) for the identification of *Nocardia* spp. The identification rates at the species level can be improved by using repeat or new extracts [9].

#### 3.1.4. Yeast Identification

Most yeast species spectra that cause infections in humans have been included in the commercial database, and evaluations of the identification of yeasts have shown good performance [71,76,77]. In a study by Zhang et al., the performance of the VITEK MS v2.0 system for the identification of yeast isolates collected from patients with invasive fungal infections in the National China Hospital Invasive Fungal Surveillance Net 2011 program was evaluated. In that study, a total of 1243 isolates representing 31 yeast species were analyzed, and 97.3% of the isolates were correctly identified to the species level. Based on these results, a testing algorithm combining the VITEK MS system with selected supplementary ribosomal DNA sequencing was developed, which could be practically implemented in strategic programs for fungal infection surveillance [10]. Based on this program, Wang et al. conducted a comprehensive evaluation of the Bruker MS (v3.1) and VITEK MS (v2.0) systems, with the latter system accurately identifying 95.4% of the isolates (2559/2683) and the former system accurately identified 98.8% (2651/2683) of the isolates [78]. In that study, neither MALDI-TOF MS systems differentiated between *Meyerozyma caribbica* and *M. guilliermondii* among the *M. guilliermondii* complex isolates [78]. Autof MS 1000 (Autobio Diagnostics, Zhengzhou, China), a commercial MALDI-TOF MS system, has been available for routine pathogen identification in many clinical laboratories in China since 2018, and its accuracy (v2.0.18) was evaluated for the identification of 1228 isolates representing 14 different yeast species within five closely related species complexes compared to the VITEK MS (v3.0). For *Candida albicans* complex, *C. glabrata* complex isolates, *C. parapsilosis* complex, and *C. neoformans* complex, the accuracy of the identification rate to the species level was 99.4% vs. 96.3%, 98.9% vs. 94.7%, 99.0% vs. 79.1%, and 99.4% vs. 95.2% when using the Autof MS 1000 and VITEK MS, respectively. In addition, the Autof MS 1000 and VITEK MS systems both performed well for the identification of *C. auris* [4]. Therefore, MALDI-TOF MS is valuable for the routine identification of yeast species. Wang et al. reported that different culture media affected the identification accuracy of the Bruker Biotyper MALDI-TOF MS and VITEK MS systems for identifying *C. tropicalis* [79]. Further research is required to determine the effect of different culture media on microbial identification.

#### 3.1.5. Filamentous Fungi Identification

For a long time, identifying filamentous fungi has been a challenge in clinical microbiology laboratories. The traditional identification methods of filamentous fungi, which rely primarily on morphology and require experienced staff, are time-consuming [80,81]. Molecular technology, which is the gold standard for identification, has solved most of these problems. However, for filamentous fungi, molecular identification often requires sequencing multiple gene loci, which is expensive [81]. Fortunately, with the application of MALDI-TOF MS in clinical laboratories, the identification accuracy of filamentous fungi has improved (Table 1), shortening the identification cycle. A recent study found that the correct identification rate of MALDI-TOF MS for *Aspergillus* filamentous fungi was more than 95% [82]. Another study showed that the MALDI-TOF MS system correctly identified 98.8% (1094/1107) to the species level [83]. In addition, for non-*Aspergillus* filamentous fungi, Li et al. observed 57.7% (30/52) correct identification to the species level by MALDI-TOF MS [11]. Peng et al. evaluated 123 strains of filamentous fungi (24 species), and Microflex LT correctly identified 65% (80/123) of the isolates to the species level and 92.7% (114/123) to the genus level [84]. Furthermore, Pinheiro et al. found that the correct identification rate of 70 clinical isolates and 20 environmental isolates identified by VITEK MS (v3.0) were 63.5% (47/74) to the species level and 82.2% (74/90) to the genus level. [81]. Several in-house supplementary databases have avoided the limitations of commercial databases. Gautier et al., Lau et al., and Becker et al. in their respective studies identified 347, 152, and 472 fungal species, respectively [80,83,85]. The identification rate to the species level with clinical strains was 98.1% (257/262, score of ≥ 1.90), 87.9% (370/421, score of ≥ 2.0), and 85.6% (334/390, score of > 1.70). The accuracy of the species identification rate by using MALDI-TOF MS was increased by using in-house supplementary databases. In addition, the type of culture medium and culture time also affect the identification of filamentous fungi. Coulibaly et al. reported that the MALDI-TOF MS spectra of a strain cultured at different times are different [86]. Li et al. proposed that the front hyphae from young colonies could improve the identification accuracy of non-*Aspergillus* filamentous fungi [11].

#### 3.1.6. Virus Identification

Many studies have demonstrated that MALDI-TOF MS can be used to identify a variety of viruses, such as human papillomavirus (HPV), enteroviruses, and influenza viruses, from clinical specimens (Table 1). Sjöholm et al. assayed a total of 882 fluid samples, including conjunctival fluid, wound secretions, blisters, plasma, serum, and The consistency rate of MALDI-TOF MS using the reference method was 95.6%, and the specificity was 98.0% [19]. Cai et al. developed a high-throughput genotyping method using MALDI-TOF MS, in which the concordance was 80.1% (285/356) [87]. Peng et al. applied MALDI-TOF MS for the identification of enteroviruses. The agreement between the results obtained using MALDI-TOF MS and DNA sequencing was 93.4% (225/241) [88]. Furthermore, MALDI-TOF MS was applied for the identification of influenza virus types. The results showed 100% (29/29) concordance with results of molecular methods [17].

### 3.2. Identification from Positive Blood Cultures

Bloodstream infection is a life-threatening disease that requires timely diagnosis and precise treatment. Compared to traditional methods, direct identification of pathogenic microorganisms from positive blood culture broth by using MALDI-TOF MS is quicker and less labor-intensive. The key to this process is to differentiate microorganisms from blood broth. Several commercial and laboratory-developed protocols have been reported to date. Owing to the lack of standardization techniques, the overall performance of these methods varies greatly, ranging from 60% to 100% at the species level [20,44]. However, direct comparisons between these studies are difficult, because the identification rates can be affected by factors such as the pre-treatment method, microorganism distribution, and the cut-off value for species identification. In a previous study, Zhou et al. developed an in-house protocol for direct MALDI-TOF MS-based identification of organisms in positive blood cultures [89]. The data showed that the in-house protocol exhibited higher performance for gram-negative bacteria than for gram-positive bacteria, shown by Zhou et al. [89]. However, Zhou et al. also developed an in-house saponin-based extraction method for the direct identification of organisms in VersaTREK-positive blood cultures by using a Bruker Biotyper MALDI-TOF MS system (MALDI Biotyper). The identification rate of these protocols was 90.6% (184/203) and 82.3% (167/203) at the genus and species levels, respectively [21]. In addition, using the MIXED method, which is recommended by the manufacturer for suspected mixed organisms, two species were identified in more than half of the polymicrobial blood cultures [89].

Apart from direct pathogen identification, the potential of MALDI-TOF MS for antimicrobial susceptibility testing has also been explored [23]. Johansson et al. applied MALDI-TOF MS to screen for *cfiA*-positive *B. fragilis* strains directly from blood culture bottles with various ertapenem minimum inhibitory concentrations, yielding results within 3 h [90]. Oviano et al. expanded their use to detect carbapenem-resistant gram-negative bacteria causing bloodstream infections [91]. A MALDI-TOF MS-based assay was established by measuring the hydrolysis of imipenem in blood cultures spiked with *Pseudomonas aeruginosa*, *Acinetobacter baumannii*, or Enterobacteriaceae producing different carbapenemases [92]. The analysis was performed using MBT Compass STAR-BL module software (Bruker Daltonics), automatically providing a result (sensitivity or resistance) based on the degree of hydrolysis of the antibiotic. The assay achieved 98% sensitivity and 100% specificity after 30 min of incubation, and both values increased to 100% after 60 min of incubation [92]. However, performing antimicrobial susceptibility testing directly from blood cultures with MALDI-TOF MS is impossible for most laboratories. Further research is required to achieve this goal; however, the potential applications of MALDI-TOF MS are promising for use in all aspects of blood culture testing.

### 3.3. Identification Directly from Patient Urine or CSF

For the identification of urine samples, positive samples were measured using flow cytometry of routine urine tests, and many bacteria were suitable for direct identification. İlki et al. reported a microorganism identification rate of 91.8% (484/538) for MALDI-TOF MS, directly from urine samples [22]. Other studies revealed that approximately 10^4^–10^6^ CFU/mL of bacteria in urine could be detected by MALDI-TOF MS [93,94,95]. Li et al. reported that MALDI-TOF MS combined with UF1000*i* could be used for rapid identification within 1 h. The accuracy of direct identification was shown to be 86.42% (229/265) of single-microorganism samples [28]. However, similar to the identification of positive blood samples, identification directly from urine by MALDI-TOF MS is more effective for identifying single bacterial samples, whereas the effect on > 1 bacterial species is poor.

Although the rapid identification of urine samples by MALDI-TOF MS is of great significance to patients, several associated problems remain. The identification rate of gram-positive bacteria is higher than that of gram-negative bacteria [96]. Several studies have shown that large amounts of urine are required, with several tests requiring 10 mL, which may be difficult to obtain from children [91,97,98].

Compared with urethral infection, CSF infection has a greater impact on patients, and rapid diagnosis and treatment are of great significance. Currently, there are few studies describing the use of MALDI-TOF MS to detect microorganisms in the CSF. One case of bacterial meningitis caused by *Klebsiella pneumoniae* was directly identified using MALDI-TOF MS [99]. Similarly, Hartmeyer et al. directly identified *S. pneumoniae* by using MALDI-TOF MS in CSF samples [100]. Torres et al. showed the direct identification of biomarkers in CSF samples from patients with enteroviral meningitis by using MALDI-TOF MS [29]. Furthermore, Bishop et al. reported that MALDI-TOF MS effectively detected gram-negative rods in smear-positive CSF samples; however, it was not suitable for identifying gram-positive bacteria [45]. In conclusion, with further study of the role of MALDI-TOF MS in CSF identification in the future, MALDI-TOF MS may be used more extensively.

### 3.4. Specific Biomarker Discovery for Detecting Antibiotic-resistant Strains and High Virulent Strains

If a specific protein in pathogenic bacteria is related to antimicrobial resistance, the quality peak of the protein can be detected by MALDI-TOF MS to distinguish sensitive and drug-resistant pathogens. MALDI-TOF MS can identify methicillin-resistant *Staphylococcus aureus* (MRSA) and methicillin-susceptible *S. aureus* (MSSA) by detecting small peptides of phenol-soluble modulin (PSM)-*mec*. The presence of the 2415 ± 2.00 *m/z* peak can predict *mecA* carriage in *S. aureus*. However, because of its low sensitivity, it is easily affected by strain differences and other factors. At present, the strains identified by MS with small peptides of PSM-*mec* are MRSA; however, strains that cannot be identified by MS as MSSA [101,102].

An extended-spectrum class C beta-lactamase from *A. baumannii* was recently identified (*m/z*~40,279) as a biomarker for carbapenem resistance. The sensitivity and specificity of the method were 96% and 73%, respectively, compared with the microdilution imipenem susceptibility testing method [103]. An 11,109-Da peak was identified by MALDI-TOF MS to distinguish the protein patterns of *Klebsiella pneumoniae* carbapenemase (KPC)-producing and non-KPC-producing *K. pneumoniae* strains [104]. A previous study revealed that the 11,109-Da peak is a cleavage product of a hypothetical protein named p019 [105]. However, as p019 polypeptide was absent in a subset of *bla*_KPC_-harboring plasmids, negative results should also be confirmed by other tests. The identification of possible markers associated with drug resistance by MALDI-TOF MS has also been investigated in anaerobes. *cfiA*-positive *B. fragilis* can be distinguished from *cfiA*-negative strains by a set of peak shifts in the range of 4000–5500 Da [106].

The screening of characteristic peaks related to drug resistance can detect the pathogen associated with drug resistance; however, the methods described above are specific for one resistance mechanism. Owing to its low sensitivity, the method must be optimized in the future to improve identification accuracy further.

Serotype VI Group B *Streptococcus* (GBS) is an important invasive pathogen, and Li et al. found that the protein peak of 6251 Da appeared in most serotype VI GBS (20/24, 92%), and the protein peak of 6891 DA was observed in most serotype III (15/18, 83%) and Ib (19/23, 83%) strains. Latex agglutination methods combined with MALDI-TOF MS may quickly identify GBS serotypes [107].

Among the *Clostridioides difficile* isolates, the hypervirulent RT027 isolates caused large-scale outbreaks in North America and European countries. Detection of the hypervirulent RT027 strains based on the absence of a peak at 6654 *m/z* and the presence of a peak at 6712 *m/z* by MALDI-TOF MS presented a sensitivity of 100%, specificity of 91.7%, and positive predicted value of 95% [24]. However, the major clone spreading in Asian countries was the hypovirulence strain, mainly ST37 and the recently identified ST81 strains. Our previous study identified five peaks, which revealed that five markers (2691.43 Da, 2704.91 Da, 2711.93 Da, 3247.27 Da, and 3290.76 Da) could easily and reliably distinguish between clade 4 and non-clade 4 isolates, with the area under the curve values of 0.991, 0.997, 0.973, 1, and 1, respectively [25].

## 4. Current Limitations of Microorganism Identification by MALDI-TOF MS

At present, MALDI-TOF MS can correctly identify most pathogens; however, the identification accuracy of some pathogens with similar mass spectra by MALDI-TOF MS is low. These are some strains of complex bacterial groups, including *A. calcoaceticus-A. baumannii* complex, *Enterobacter cloacae* complex, *Burkholderia cepacia* complex, and *S. mitis* complex [15,108,109,110]. In addition, *Shigella* and *Escherichia coli* cannot be accurately identified by MALDI-TOF MS because they have similar mass spectra [111]. For the *M. tuberculosis* complex too, one of the reasons that affect the effectiveness of MALDI-TOF MS is the similarity of the mass spectra. The identification accuracy of H. pylori is low because there are currently fewer mass spectra for this species in the database. With the continuous enrichment of spectral databases and improvement of spectral quality, MALDI-TOF MS will be more widely used in clinical laboratories [80,83,112].

**Table 1 microorganisms-09-01536-t001:** Performance of MALDI-TOF MS for the identification of microorganisms.

Authors, Year of Study	No. of Isolates Evaluated	Percentage (no.) of Isolates Correctly Identified by MALDI-TOF MS to the Indicated Level	
		Genus Level	Species Level
**Aerobic bacteria**			
**Gram-negative**			
Faron et al., 2015 [60]	2263	99.8% (2258)	98.2% (2222)
Kassim et al., 2017 [61]	222	95.9% (213)	87.8% (195)
**Gram-positive**			
Rychert et al., 2013 [62]	1146	95.5% (1094)	92.8% (1063)
Kassim et al., 2017 [61]	131	100% (131)	100% (131)
**Anaerobic bacteria**			
Garner et al., 2014 [65]	651	92.5% (602)	91.2% (591)
Jamal et al., 2013 [7]	274	99.3% (272)	89.1% (244)
Rodríguez-Sánchez et al., 2016 [66]	295	97.0% (286)	85.8% (253)
**Yeasts**			
Wang et al., 2016 [78]	2683	Not applicable	98.8% (2,651)
Yi et al., 2021 [4]	1228	99.2% (1,218)	89.2% (1,095)
Maldonado et al., 2018 [113]	201	Not applicable	92.5% (186)
**Filamentous fungi**			
Becker et al., 2014 [80]	390	Not applicable	95.4% (372)
Gautier et al., 2014 [83]	1107	Not applicable	98.8% (109)
Li et al., 2020 [11]	52	Not applicable	57.7% (30)
***Mycobacterium* species**			
Wilen et al., 2015 [50]	157	Not applicable	89.2% (140)
Rodrigues-Sanchez et al., 2015 [46]	125	88.8% (111)	68.8% (86)
Chen et al., 2013 [71]	102	87.3% (89)	62.8% (64)
**Virus**			
Sjöholm et al., 2008 [19]	882	Not applicable	95.6% (843)
Cai et al., 2019 [87]	356	Not applicable	80.1% (285)
Peng et al., 2013 [88]	241	Not applicable	93.4% (225)

## 5. Conclusions

MALDI-TOF MS has been widely used in clinical microbiology for a dozen years. Nowadays, MALDI-TOF MS can identify common microorganisms to the species level. Moreover, MALDI-TOF MS systems can technically identify some bacterial subspecies [114,115,116,117]; however, with optimizing methods and technological innovation, MALDI-TOF MS still has more application growth in this field. The reliability and accuracy of MALDI-TOF MS have been analyzed in many studies, and the spectrum database is vital for identification. In the future, MALDI-TOF MS is expected to play a more critical role in promoting the development of clinical microbiology because it is fast, accurate, sensitive, and has a high throughput.

## Data Availability

The data presented in this study are available on request from the corresponding author.

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
