# Peer review of "Matrix-Assisted Laser Desorption/Ionization Time of Flight Mass Spectrometry (MALDI-TOF MS) Analysis for the Identification of Pathogenic Microorganisms: A Review"

_microorganisms, 2021, doi:10.3390/microorganisms9071536_

Round 1
Reviewer 1 Report
Xin-Fei Chen and the co-authors present here a review entitled: Matrix-Assisted Laser Desorption/Ionization Time of Flight Mass Spectrometry (MALDI-TOF MS) for the Identification of Pathogenic Microorganisms. The review is very hard to read it has no flow, some sentences are misleading and/or not clear. The aim of this review was to summarize pathogenic identification, sample preparation and current problems using MALDI-TOF MS in clinical microbiology labs. However, in my view this was not obtained here, it was not an up-to-date summary. References are missing for many statements.
Abstract:
Line 24: decades – As MALDI-TOF MS was launched in 2010 as a diagnostic tool and previously was only for research. Decades seem a bit too much.
Introduction:
Line 39: the same as above
Lines 43-45: “MALDI-TOF MS is not limited to identifying …”. What does that mean? Please rephrase this sentence.
Line 46: I would like to see a reference to this statement at the end of the sentence.
Line 49: same as above reference is missing.
2.1. MALDI-TOF MS basic overview:
Lines 58-66: Very difficult to read and unclear paragraph. Also, m/z is used here for the first time but not explained e.g. mass to charge ratio.
2.2. Sample preparation
Lines 73-74: reference needed
2.2.1. General preparation with laboratory culture strains
Line 89: reference
2.2.2. Liquid samples
Line 93: reference missing
2.2.3. Mycobacterium and Nocardia spp.
Lines 122-132: Again, very difficult to read and unclear paragraph.
2.2.4. Filamentous fungi
Lines 134-135: What is the home-brew method? Do you have a reference to this?
2.3. Classification and Identification of Microorganisms
2.3.1. Identification from culture plates
Line 145: at the end of this sentence I would like to see a reference to this statement.
Line 147: the same as above
2.3.2. Anaerobe identification
Line 182: “Rodríguez-Sánchez” add “et al.”
Line 189: “Bacteroides” should be written Italic
2.3.3. Yeast identification
Line 202: “In this study” is misleading, I would change it to “in that study”.
Line 210: the same as above
Line 216: “VITEK MS (v 3.0).” space
2.3.4. Identification from positive blood cultures
Line 236: “In our previous study” as this is a review I would not use this line.
Line 251: References throughout the manuscript: add the note to the text together with the citation “Ovian et al. [54].”
2.3.5. Identification directly from patient urine or CSF
Line 287: this sentence is not finished
2.3.6. Mycobacterium identification
Lines 293-323: This whole section is very difficult to read and not clear. Please rephrase it and add references where appropriate and up to date.
2.3.7. Filamentous fungi identification
The same as above
2.3.8. Specific biomarker discovery for detecting antibiotic resistant strains and high virulent strains
The same as above. It is not clear how MALDI is used for resistance testing. There are publications on sensitivity/susceptibility testing using MALID too.
- Current Limitations of Microorganism Identification by MALDI-TOF MS
In general, difficult to read and understand. Especially lines 402-404: makes no sense
Reviewer 2 Report
The authors presented an interesting review on MALDI-TOF pathogen identification in clinical settings. Because the manuscript covers many of the practical aspects of the technology, this review may benefit the readers. However, some major points should be overcome to consider publication.
Major concerns.
Throughout the manuscript, the word identification seems to be dedicated to species-level identification, but it is not clearly stated. MALDI identification in clinical level includes genus-level, species-level, and subspecies-level, although identification softwares tend to show the results in terms of species. The authors should state in the manuscript that the word identification means species-level as in most cases in clinical situations.
Also, I recommend the authors to state that usual MALDI systems can technically identify some bacterial subspecies (Ruiz-Moyano et al. Food Microbiol 2012, Fangous et al. J Clin Microbiol 2014, Nie et al., J Clin Microbiol 2015, Dekio et al. Int J Syst Evol Microbiol 2015).
2.1. MALDI-TOF MS basic overview. Although all the proteins are ionized in the process, MALDI-TOF can detect only ions with the m/z values less than 30 kD (Shimadzu system) or 20 kD (Bruker system). Therefore, the spectra (in clinical settings) only include only a small part of these. Please amend the text based on this.
2.3. Classification and Identification of Microorganisms. The paragraphs in this part includes: 2.3.1. Identification from culture plates, 2.3.2. Anaerobe identification 2.3.3. Yeast identification, 2.3.4. Identification from positive blood cultures, 2.3.5. Identification directly from patient urine or CSF, 2.3.6. Mycobacterium identification, 2.3.7. Filamentous fungi identification, and 2.3.8. Specific biomarker discovery for detecting antibiotic resistant strains and high virulent strains. The order and subtitle headings seem awkward in several points. First, for 2.3.1, the authors seem to include aerobic bacteria here, but it is not clearly stated as so. Second, subheadings about identification procedures and those about pathogenic differences are mixed. I think these are more efficiently composed, for example: 2.3. Idenfication procedures in different occations, 2.3.1. Identification from culture plates, 2.3.4. Identification from positive blood cultures, 2.3.5. Direct identification from patient urine or CSF, 2.4. Identifications in various microorganisms, 2.4.1. Aerobic bacteria, 2.4.2. Anaerobic bacteria, 2.4.3. Mycobacterium, 2.4.4. Yeast, 2.4.5. Filamentous fungi, and 2.4.6. Specific biomarker discovery for detecting antibiotic resistant strains and high virulent strains.
There are a few different platforms for MALDI-TOF identification. If possible, the authors should refer to the features and differences of these, at least, Bruker, Shimadzu, and Biomerieux systems.
Round 2
Reviewer 1 Report
This review has been extensively improved and I recommend it for publication.
Reviewer 2 Report
I feel the manuscript is much improved and it is nearly suitable to publish, however, I see some minor points to correct.
Minor points.
Line 71. Not 30,0000 Da but 30,000 Da.
Line 430. "Usually” should be “Moreover”
Author Response
Please see the attachment

This manuscript is a resubmission of an earlier submission. The following is a list of the peer review reports and author responses from that submission.